



# 1  X-ray CT analysis of pore structure in sand

**Toshifumi Mukunoki[1], Yoshihisa Miyata[2], Kazuaki Mikam[3] and Erika Shiota[1]**
[1] X-Earth Center, Graduate School of Science and Technology, Kumamoto University, 1-39-2
Kurokami Kumamoto-city, Kumamoto, JAPAN
[2] Department of Civil and Environmental Engineering National Defense Academy, 1-10-20
Hashirimizu, Yokosuka, JAPAN
[3] Japan Oil, Gas and Metals National Corporation, Toranomon Twin Building 2-10-1 Toranomon,
Minato-ku, Tokyo, JAPAN
Corresponding author to: Toshifumi Mukunoki (mukunoki@kumamoto-u.ac.jp)
ABSTRACT
The development of a micro-focused X-ray CT device enables digital imaging analysis at the
pore-scale. The applications have been diverse, for instance, in soil mechanics, geotechnical and
geoenvironmental engineering, petroleum engineering, and agricultural engineering. In particular,
imaging of the pore space of porous media has contributed to numerical simulations for single and
multi-phase flow, or contaminant transport, through the pore structure as three-dimensional image
data. These obtained results are affected by the pore diameter so it is necessary to verify the image
pre-processing for image analysis, and validate the pore diameters obtained from the CT image data.
Besides, it is meaningful to produce the parameters in a representative element volume (REV) and
significant to define the dimension of REV. This paper describes the underlying method of image
processing and analysis and discusses the physical properties of Toyoura sand for the verification of
image analysis based on the definition of REV. Based on the obtained verification results, pore
diameter analysis can be conducted and validated by the comparison of the experimental work and
image analysis. The pore diameter was deduced by Laplace's law and the water retentively test for
the drainage process. The referenced results and perforated-pore diameter proposed originally in
this study, called the voxel-percolation method (VPM), are compared in this paper. The paper
describes the limitation of REV, the definition of pore diameter, and the effectiveness of VPM for
the assessment of pore diameter.
Key words: Pore diameter, Image analysis, REV, percolation, X-ray CT



## 1. INTRODUCTION

The estimation of pore dimensions and pore networks in soil is one of the most important studies to evaluate the mechanical and hydrodynamic properties for soil science, soil mechanics, geotechnical and geoenvironmental engineering, and petroleum engineering [Carman (1939), Brooks and Corey (1964), Topp and Miller (1966), Bear (1972), Mualem (1976), Chatzis et al. (1983), Dullien (1992), Helming (1997), Chanpus (2004), Culligan et al. (2006), Riyadh (2007), Gharbi and Nlunt (2012) ]. In fact, it is difficult to define pore dimensions in grains because the pores are surrounded by grains and are thus not isolated. Figure 1 illustrates a pore in spheres. As shown in Figure 1(a), five spheres surround one pore such that the pore should be defined by nine contacting points. However, the pore space is not closed by the five spheres. The shape of soil particles is not spherical, but rather a complicated shape so the pore dimension is able to be defined based on assumption only. Figure 2 shows the X-ray CT image of a grain sample in two dimensions. The X-ray CT shows the spatial distribution of density, which enables the soil particles and pores to be distinguished. Locally, the longest and shortest length of the pore, as shown in Figure 2 (a), can be measured by using software for image analysis. However, it is partial property of the pore, and the required information is at least a property in representative volume. It should be required not to measure individual pore dimensions by using software, but to estimate them by using a systematic method. Moreover, complicated pores have an aspect ratio as shown in Figure 2, so the discussion of connectivity of pores will be required for the study of the hydrodynamic issue in soils. The challenge of this paper is to propose an evaluation method for the pore dimensions.

Here shall we look back the current technique on the pore analysis. The most popular method for measuring pores in soil is    the indirect methods of the mercury intrusion technique (MIT) or the air intrusion method (AIM). These methods are based on the concept that the pore structure assumes a straight tube. Thus, the three-dimensional pore network is not an issue. Through recent developments of the scanning electron microscope (SEM) and non-destructive testing methods such as computed tomography (CT) and magnetic resonance imaging (MRI) the pore structure in soil can be measured directly. In particular, CT is applicable by using rays, for example, sound, ultrasound, x-ray and gamma ray, so that the pore structure of various engineering materials [Otani and Obara (2003), Desrues et al. (2006), Alshibli and Reed (2010) and Cnudde and Bernard (2013)] can be scanned. Additionally, advanced CT has been developed to scan in the micro scale [Altman   et al.(2005), Wildenschild et al. (2002), Wildenschild et al. (2005), Wildenschild et al. (2005), Mukunoki et al. 2010, Higo et al. (2011), Wildenschild and Sheppard (2013), Andrew et al (2014),





Andrew et al. (2015) and Taylor et al. (2015)]. To evaluate a structure consisting of a great number
of pores, a suitable image analysis method is required.
The most popular method to evaluate the porosity of porous materials from CT is based on a
statistical assumption. Parameters of the distribution function based on CT data can be determined
by an optimization technique [e.g. Kato et al. (2014) and Mukunoki et al. (2014)]. The accuracy of
these methods depends on the selection of distribution functions based on the CT data. The required
number and type of functions are still under discussion and there may be a number of solutions to
these issues.
This paper discusses the evaluation method of pore structure of sand from micro focused CT scan
data. In this paper, authors distinguish pore from pore structure. In the first part of the paper, the
authors propose the application of the mathematical morphology method for estimating the pores of
sand. By showing the analysis results of simple subjects, the usefulness of proposed method will be
validated. Next, the importance of the selection of a representative element volume (REV) is
discussed for estimating the grain size distribution and the averaged pore index, such as the porosity
and specific surface. The authors show there is an optimum REV in this analysis. Based on the
above fundamental examination to treat CT data, the authors propose a voxel-percolation method
(VPM) to evaluate the pore structure of sand. The estimated results are compared with a water
retention curve test, and the effectiveness of proposed method is described. It concluded that the
required resolution to evaluate the pore structure is almost equivalent to that for a pore.    The final
objective of this research is to develop a general method for soils. As a primary research, the
evaluation of sand will be treated in this paper because it is natural material and has a uniform grain
shape.
**2. X RAY-CT SCAN**
Table 1 lists the specifications of the micro-focused X-ray CT scanner (TOSHIBA TOSCANNER
32300 FPD) installed at the X-Earth Center at Kumamoto University in 2010. In general,
360-degree radioscopic image data, for an inspection object placed on a sample table, is obtained
using an X-ray image intensifier by turning the table while irradiating the object with X-rays. This
radioscopic image data is then used in reconstruction calculations, which result in cross-sectional



images. Because the generated X-ray beam is a polychromatic beam with a wide range of
frequencies, corrections are made for the beam-hardening effect. Radioscopic images alone are not
sufficient to accurately represent internal components that have a complex structure. Tomographic
reconstruction allows the detection of fine flaws, foreign matter, separation, and other phenomena.
State-of-the-art technology allows for high-precision and high-speed inspection, permitting new
applications in various fields. Figure 3 shows an illustration of the internal view of the
micro-focused X-ray CT scanner. The detector is a flat panel detector (FPD), which enables
three-dimensional scanning with a cone-shaped X-ray beam. The scan speed depends on the
scanning conditions. The sample was placed on the scan table and scanned with the cone-shaped
X-ray beam. During scanning, the scan table was rotated to obtain a 360° scan. A back projection of
the X-ray attenuation was detected on the FPD. The X-ray CT images obtained were free from the
ring artifact normally seen in CT images because the scanner applied a filter function to reduce this
during the image reconstruction process.
Table 2 lists the scan condition selected for this study. An of x-ray tube voltage of 60 kV and a
current of 200 μA were chosen, and the focus-to-center distance (FCD) was defined as 24.7 mm;
hence the dimension of one voxel is 5×5×5 μm in this study. In general, the scan condition depends
on the target of study and the voltage, current and FCD should be variable. The scan conditions
selected in this study are not absolute conditions, and each user has to find the best condition to
observe the target in each study. Figure 4 shows the photograph of the scan setup in the
micro-focused X-ray CT room. To obtain high-resolution images, the specimen must be in close
proximity to the X-ray tube, as shown in Figure 4. The soil tested was Toyoura sand with a dry
density of 1.57 t/m$^3$ (i.e., the porosity was 0.41). The sample was well packed into an acrylic mold
with a diameter of 10 mm and a thickness of 1 mm. If the diameter of the specimen is greater than
10 mm, for example 20 - 30 mm, the center of the specimen is too far from the X-ray tube and
high-resolution CT images cannot be obtained because of the loss of focus distance between the
X-ray tube and the center of the specimen.
**3. IMAGE PROCESSING OF A PORE IN SAND BASED ON X-RAY CT DATA**
**3.1 Outline of image processing of a pore in sand**
There may be some methods of treating image processing of a pore in sand from X-ray CT data. In
this    paper,    the    applicability    of    mathematical-morphology    is    discussed.    The





mathematical-morphology shows the complicated form by using element whose dimension is
known [Soille (2003)]. The basic operations are available in many image analysis software
packages [Luis et al. (2005)]. The pore has a complicated form so this concept may be useful for
evaluation.
The basic process for image analysis in this study is as follows:
1) Image segmentation to create a binary image from an original image (3.1.1);
2) Determination of pore diameter using a granulometric method based on mathematical

9        morphology (3.1.2);

3.1.1 Image segmentation
As the first step of image analysis, the pore and grain are identified from CT-data. For the operation,
the image segmentation method developed by [Otsu (1979)] was used because the Toyoura sand
tested showed two distinct peaks in this study. Two peaks of CT values indicate the two phases of
particles and pore space, respectively. During this process, the binary data is set such that the grain
is black and the pore is white. By determining the voxel number for each region, the averaged index
showing the pore, such as the void ratio, can be evaluated. However, by this process, the
distribution of the pores according to size cannot be evaluated.
3.1.2 Granulometric method
In the second step of image analysis, for each pore which is identified by binary data, the shape is
evaluated by using mathematical morphology. Based on this concept, unit element B belongs to a
pore X. Then the unit element in X, $B_x$, can be written by following equation.
$$B_X = B \cup X \qquad (1)$$
It is likely that the shape of unit element is a square or a circle with a symmetric shape. In this study,
the target subject is a pore and the interest is its size. Obtaining the image of a pore from the X-ray
CT scanner, the dimensions of the pore can be shown by using the unit element, because dimension
of the unit element can be regulated. In this study, the unit element of a sphere is used, thus the
element is called the sphere element in this paper.
Figure 5 illustrates the sphere element. In this study, thirteen sizes of sphere element are used. In



this figure, the first three elements are shown. The number of the maximum sphere element used as
a radius is r=13 in this study. The smallest element is one voxel (Figure. 5(a)). Elements with a
lower sphere number do not appear exactly spherical in shape. However, the greater the number of
voxel for diameter of structural element, the more it spherical it becomes, as shown in Figure 5. The
voxel number as a diameter of a sphere element, D can be defined as per following equation:
$D=2r+1 \quad r \geq 0$                (2)
where r is a voxel number from the center of the sphere element. As shown in Figure 5, when r is 1,
which is the minimum number, d is 3; therefore, the diameter of sphere element is 3. The diameter
increased based on the center inclusion concept, so the diameter of the sphere element should be
always an odd number. The more r increases, the more the shape of the sphere structure l becomes
spherical. For the generation of a sphere element with different dimension, Image Tool Kit called
ITK [Luis et al. (2005)] was used.
The granulometric method can recreate the complicated pore space by overlapping sphere elements
with several different diameters. Initially, the smallest sphere element, i.e., one voxel, should be
applied to the analysis area, which is a pore space, and it should occupy the entire pore space. Then,
the next sphere element with a diameter of 3 voxels, as shown in Figure 5, is applied same area of
pore space, but the next sphere element cannot occupy the entire pore space. Likewise, the entire
pore space is scanned by each sphere element and the more complicated the pore shape, the more
sphere elements with different diameter will be required. In the granulometric method, the sphere
element is overlapped partially; hence, the distribution of pore diameter indicates that the
non-overlapped part of sphere element is evaluated by the voxel number. In addition, the
summation of the overlap ratio of the sphere element at each step can be used to evaluate the pore
volume and saturation degree by the voxel percolation method, explained in the following section.
**3.2 Verification of image processing for simple objects**
Figure 6(a)-(e) explains the features of the granulometric method. Figure 6(a) and (b) illustrate a
circle and a square rotated 45° in the dimension of 200 x 200 (i.e. the total voxel number is 40000),
and Figure 6(c) prepares the circle and square in the dimension of 400 x 200 (i.e. the total voxel
number is 80000). Let us define the space other than the circle and square as pore space. It is noted
that authors refer to a sphere element as a circle element in this section because the target subject is





drawn in two dimensions. At the end of the image analysis using the granulometric method, herein
referred as granulometric image analysis (GIA), authors can count the number of voxel involved in
each sphere element. All images classified by each sphere element, as shown in Figure 6, were
overlapped in the order of the diameter of the circle element. The number of voxel for a minimum
circle element is equal to the number of pore structure in total. Hence, the number of voxel for
circle element greater than the minimum circle element should be less than the number of pore
spaces. Finally, all circle elements are overlapped due to the radius of the circle element and the
total number of voxel is same as that of the pore space, as shown in Table 3; and then, it can be
expressed by the following set equation;
$S = \sum_{i}^{n} B(r_i) - \sum_{i}^{n} \{B(r_i) \cap B(r_{i+1})\}$   $i$=0, 1, 2….$n$   (3)
where $S$ is the number of voxel as area/volume counted by GIA, $B(r_i)$ is area/volume of the
circle/sphere element with a radius from $i$ to $n$. When $i$ is $n$, $S$ is equal to the target area/volume.
Even if the shape of the pore space is complicated, as in Figure 6(c), GIA can estimate the voxel
number in total. Note that each circle element labeled and its location were also recoded; hence, the
spatial distribution of the overlapped circle element can be visualized as shown in Figures 6. Here,
it should be recognized that the diameter of the circle element cannot become a pore diameter
directly. Hereby, a definition of pore diameter is required.
If the diameter of sphere element is equivalent to the pore diameter, its pore geometry must
be a set of parallel lines or a rectangular shape. The area in dotted lines, as shown in Figures 6 (d)
and (e), was analyzed by GIA and both areas were found to be 19801. For latter case, the two
artificial images in Figures 6(d) and (e) can provide an interesting discussion. Certainly, GMI
estimates the width in the area of the diamond; despite the fact that the definition of width is vague,
GMA produced Figure 6(d). Even if the target image is rotated by 45° , the same results should be
obtained. GMA searches the minimum pore space first and then, the sphere element with the
diameter due to order of odd number becomes larger. The point is the sphere element evaluates the
space at four corners, as shown in Figure 6(e). This result indicates the important feature of this
image analysis; in short, the sphere element finds small pore spaces and evaluates the diameter by
the part remaining after the overlapping process. This behavior, whereby the sphere element finds
the small space, is similar to capillary behavior in porous materials such as sand.
**3.3 Selection of appropriate Representative Element Volume (REV) for analysis of sand**



The representative element volume (REV) of the subject should be discussed to estimate the
geometrical properties of sand by image analysis. In this study, the porosity and specific surface of
Toyoura sand were evaluated, and their REV was assessed. Detailed steps can be referred from
[Fujiki et al. (2014)]; hence, the concept is only introduced in this paper as follows:
1)    Sub-sampling region is defined by the authors;
2)    The porosity and specific surface are calculated from the CT-image;
3)    Their mean-value and standard deviation of the porosity ($n$) and specific surface ($S_{sp}$)

8          defined by the following equations are calculated; and then,

$$n = \frac{V_{pore}}{V_t} \qquad (4)$$
$$S_{sp} = \frac{S_{ps}}{V_t} \qquad (5)$$

14        where $V_{pore}$ is a volume of pore, $V_t$ is the total volume of specimen and $S_{ps}$ is mean-surface

15        of grains obtained from CT image.

4)    The process from the above items 1 to 3 is continuously repeated due to the enlargement of

17         the calculation region until reaching the relative standard deviation (RSD) which is defined

18         by the equation expressed as the standard deviation divided by mean-value, isless than 1%.

20        Figures 7(a) and (b) show the two-dimensional images of particles of Toyoura sand in each

square dimension. The dimension of the CT image of Figure 7(a) is 1024 x 1024; however, if the
extraction of the cubic area from this CT image is required, the maximum voxel number to be used
is 700, as shown in Figure 7 (a). These images can be obtained from binary images following the
image segmentation process. In this study, the Ohtsu method was applied to create binary images.
For each three-dimensional image, the porosity and specific surface were evaluated to validate the
representative volume. The measured porosity of the specimen was 0.44, and the analyzed porosity
was 0.431, so both values were a good fit. Figure 8 presents the evolution of the relative standard
deviation (RSD) for the porosity and specific surface with a sub-sample size. A theoretical
decreasing behavior is observed for both cases. The behavior observed for the two cases generally
results in close values. The difference between the porosity and the specific surface is not
significant for the tested materials but, generally, the REV should depend on the medium property.



Figure 9 shows the distribution curves of the grain size obtained from the sieving test and image
analysis. The grain size distribution curve by image analysis can be obtained from Image J, which is
provided using the function of object counter. With the exception that the image analysis area is
cubic of 100 voxels, the grain size distribution obtained from the CT image has a good fit to the
results of the sieving test. Figure 10 shows the relationship between the image analysis area and the
particle diameters analyzed. The subscript number D means the percentage value finer, by weight,
obtained from Figure 9. The Figure 10 results show that a cubic area of more than 300 voxels can
provide a constant grain size for each percentage finer by weight. Defining a limit for the RSD in
order to choose the size L of the REV remains an open question. The orders of magnitude are
summarized in Table 4. The effect of the size of the reference sample is not significant. This
observation supports the fact that the 300 voxel size (or 1.5 mm size) sample is larger than the REV
when one voxel size is 5 μm.
**4. ANALYSIS METHOD OF PORE-STRUCTURE**
**4.1 Pore structure analysis**
In the process of section 3.2, GIA produces pore by the overlapping of many sphere elements. In
this chapter, pore structure analysis based on GIA is described.   It is important to consider
three-dimensional continuity of the pore in analyzing the pore structure. In this paper, the pore
structure analysis method to perform vertical air-entry simulation with the imaged pore from X-ray
CT data is proposed. This method is called the voxel-percolation method (VPM) in this paper. For
instance, the number of the kind of sphere element is 13, as shown in Figure 11. For convenience,
the images in Figure 11 are drawn in two dimensions; therefore, sphere elements should be called
circle elements in the explanation of Figure 11. In order to start the percolation flow simulation, in
this study, the rule for water-drainage process should be as follows:
Step 1: The original image is binarized to pore space (white) and soil particles (black) (see Figure

26         11(a));

Step 2: The pore space is analyzed by GIA, and thus the distribution of the labeled sphere element

28         can be known (Figure 11(b));

Step 3: VPM starts to find the labeled voxel (herein as 13) of the circle element with the largest

30         radius from the corner of the defined side. As in Figure 11(c), only the area of the sphere

31         element with a radius of 13 is shown as white.

Step 4: Once VPM finds the labeled voxel of 13, it will keep painting those voxels until it

33         recognizes no continuous circle element (Figure 11(c)-(o)). Likewise, the labeled number




of the largest sphere element is scanned to the image treated in Step 2, and so only the
voxel corresponding to the largest sphere element is counted. Step 3 should be repeated
until the smallest sphere element is used;
Step 5: The results in Step 4 produce the saturation degree by the summation of the counted voxel
divided by the number of voxels of the entire pore; and
Step 6: The capillary pressure can be evaluated by using Young-Laplace's equation with the
diameter of the sphere element.
**4.2 Analyzed water retention curve**
The water retention property of a soil is a typical parameter influenced by pore structure. In this
section, the water retention curve can be reproduced by combining GIA and VPM. Based on section
4.1, the water retention curve (WRC): $h_p$-$S_r$ for the drainage process can be drawn. Figure 12 shows
the 3D image of the percolation flow. As a first step, VPM gave the distribution of the connecting
sphere elements with different size, and then, all voxels with sphere elements share the same label.
By sharing same label, the behavior of voxel seems to flow, as shown in Figure 12(a) – (f), and this
is percolation flow. Figure 13 shows the occupation ratio of the cumulative volume of the sphere
element. In fact, the occupation ratio of the cumulative volume of the sphere element countervails
the volume ratio of air in the pore structure, and so the saturation degree can be evaluated by
subtracting the cumulative volume of the sphere elements from the entire pore volume. This can be
expressed by:
$$S_r = 1 - \frac{\sum_i^n B_i - \sum_i^n B_i \cap B_{i-1}}{V_T} \qquad (6)$$
where $V_T$ is a volume of entire pore structure. The diameter of sphere element at each step
contributes to the calculation of capillary pressure head ($h_p$) by the Young-Laplace equation as
shown in eq. (7):
$$h_p = \frac{4T\cos\theta}{\gamma_w d} \qquad (7)$$
where T is the surface tension between the water and air (72.88 mN/m at 20$^{\circ}$C), $\theta$ is the contact
angle (49$^{\circ}$) , $\gamma_w$ is the density of water, and d is the diameter of the tube.
In this study, the WRC test for the drainage process could be performed so it was simulated by the



following treatments.
1) To label each sphere element categorized and to recognize the label number of the sphere
element with the maximum diameter;
2) To count the number of voxel with the label number corresponding to the maximum diameter
from the direction of air entry side, and it should be continued until the discontinuous
condition, as shown in Figure 12(a) – (f);
3) The first item yields the capillary pressure head using the Young-Laplace equation with the
substitution of the latest perforated diameter. The second item yields the saturation degree by
dividing the number of voxel not counted by the total number of pore spaces. This can be
plotted on one WRC.
4) Once the counting process on the above second item is finished, the next label of sphere
element should be checked based on same process as item 1), 2) and 3); and lastly, the WRC
can be created. In order to verify the perforated pore diameter, WRCs were obtained from the
experiment at 20 $^{\circ}$C, and image analysis was evaluated in this study.

## 4.3 Water retentively test to verify pore structure analysis

In this study, a water retentively test with a reducing elevation head method (WRT-REHM) was
selected to conduct water drainage tests because it was available to measure the moisture content of
identical specimens at different elevations head during the water drainage process. The specimen
used for WRT-REHM was identical to the scanned sample. Figures 14(a) and (b) show photographs
of the set-up used for the water drainage test system with a suction method, and Figure 14(c)
illustrates the cross-sectional view of a mold that was tested. The mold is made of an acrylic,
through which an X-ray beam could be transmitted without strong beam-hardening. The dimensions
of the mold were: height of 120 mm, inner diameter of 10 mm and a thickness of 1 mm. In order to
measure the amount of drained water, a glass syringe was used with a scale of 0.01 ml. A
membrane filter with a mean-pore diameter of 0.2 μm was placed on a glass filter with a pore
diameter between 20 and 30 μm installed on the bottom of the mold. Table 1 summarizes the
specification of the soil tested. All test procedures are listed as follows:
1) The mold was filled with de-aired water and an entire system with a syringe, tube connected
between the mold and syringe, glass filter, and membrane sheet was fully saturated in the
storage mold under vacuum condition.



2)   Toyoura sand was carefully installed in the mold filled with de-aired water.
3)   The sand specimen was left for 24 hours after the regulation of the elevation head of the syringe

3        to lead water drainage at each saturation degree, and then the volume of the syringe was

4        recorded.

4)   Items 3) and 4) were repeated until no water drainage was observed.
The entire test system was set up in the room with the installed micro-focused X-ray CT scanner,
and the temperature was controlled at $20\mp2^{o}$C. Due to a change of temperature within the range of
$20\mp2^{o}$C, the authors were concerned about the generation of condensation in the mold and extra
evaporation from the specimen surface, so the humidity in the room was regulated. In the trial test,
the authors monitored the time to reach a steady condition of the specimen through the checking of
the fluid volume in the syringe, and it was concluded that this time should be 24 hours in this test.
**4.4 Verification of pore structure analysis**
Figure 15 shows the binary 3-D image of the pore space of Toyoura sand based on the method
explained in section 3.1. In this figure, soil particles are invisible. Figure 16(a) to (e) show binarized
X-ray CT images obtained from Figure 16 in two dimensions after GIA using 13 different sphere
elements; and Figure 16 (f) is the final analysis results by overlapping each result. White represents
pore space and black represents the soil particles. Eventually, this image processing was conducted
in three dimensions so a 3D map can be obtained, as shown in Figure 17. Visually, each color
element is distributed uniquely, as shown in Figure 17. Two neighboring elements from the 3D map
are found to have neighboring colors in the color bar. That is pore size distribute continuously.
From the view point of hydraulic behavior, local velocities of pore water are always different at
each pore.

25       Figure 18 shows X-ray CT images with respect to air intrusion, obtained from the VPM

analysis in 300 voxel dimensions at each capillary pressure analyzed by equation (5) with a
diameter of a sphere element. The number of voxel count produced the saturation degree (Sr); hence,
the water retention curve (WRC) for drainage can be obtained as an image-analyzed curve. Figure
19 presents the comparison of the saturation degree obtained from Figure 18 and that from the
Toyoura sample, and thus the analyzed saturation degree was verified by Figure 19. Figure 20
shows the WRC analysis results for five voxel dimensions. Refer to Figure 5(b) to determine the
effect of voxel number on analysis results. Authors validated the effect of the voxel dimension to





WRC. In Figure 20, the test result obtained from the laboratory test is also plotted. Focusing the effect of the voxel dimension as REV, when Sr is greater than 0.8, it is observed that the lower voxel dimensions yields a 50% underestimation of the measured data to an air entry pressure as a result of decrease in Sr, remarkably. In this case, as shown in Figure 18, the voxel dimension of 100 is not sufficient to become REV for the WRC evaluation where Sr is greater than 0.8; however, there is no difference between the voxel dimensions where Sr is less than 0.5. Providing the voxel dimension is more than 300 based on all issues discussed, VPM could provide a reasonable pore diameter and the authors concluded that the diameter of the sphere element can become the pore diameter.

Despite the small change in capillary pressure between 30 and 40 cm, $S_r$ decreased from 0.9 to 0.3. This behavior indicates that mean pose size caused a capillary pressure head of 30-40 cm is mainly distributed. This behavior should be caused by sands with a value less than the uniform coefficient.

**4.5 Discussion**

The studies which require pore structure, are fluid mechanics, geoenvironmental engineering and petroleum engineering [Blunt (2001), Blunt et al. (2002), Blunt et al. (2013), Mostaghimi et al. (2013) Iglauer et al. (2013) and Muljadi et al. (2015)]. The issues of how to model migration of oil in porous media such as rocks/soils, and how to inject air for remediating contaminated soil by fuels, require that the water/oil flow in the soil quantitatively understands the pore structure [Morrow and Songkran (1981), Parker et al. (1987), Pantazidpou and Sitar (1993), Mayer and Miller (1993) and Soga et al. (2003)]. Normally, the distribution of pore diameters should be required to evaluate the water retention curve (WRC) as a hydraulic property of soils. The mercury intrusion technique (MIT), scanning electron microscope (SEM), and the air intrusion method (AIM) [Sato et al. (1992), Kamiya et al. (1996) and Uno et al. (1998)] have been used to measure the pore diameter. In general, MIT is used for the evaluation of pore size in clay, so Sato et al. (1992), and Kamiya et al. (1996) attempted to develop the method to evaluate pore size in sandy soil. Uno et al. (1998) included the moisture characteristic property in the results obtained from AIT [Kamiya et al. (1996)], and proposed the moisture characteristic curve method (MCCM). The measurement principle of AIM is similar to that of MIT and the obtained pore size is evaluated as the diameter of a pipe; however, the water contents were not measured. Uno et al. (1998) deduced the capillary pressure head using the Darcy's equation for air permeation, and then they evaluated the pore size based on a pipe model [Kamiya et al. (1996)]. WRC is composed of a saturation





degree and capillary pressure head measured by the head method with suction, or given by
Young-Laplace's law with a diameter of a pipe as the representative pore diameter. AIM gives the
statistic distribution of a pore diameter model of sandy soil as a glass tube. In short, the pore
diameter is defined as the diameter of the tube as per implicit agreement in a number of papers.
Figure 21 is a distribution curve of a perforated pore diameter for Toyoura sand and the pore
diameter deduced by the air intrusion method (AIM) proposed by Uno et al. (1998). Figure 22
presents the relationship between the image analysis area and the perforated pore diameter in this
study. Figure 22 verifies that the cubic area of more than 300 voxels can provide the constant
perforated pore diameter for each percent finer by volume. Hence, the results analyzed in a cubic of
300 voxels is compared with Uno et al. (1998). Figure 23 presents the comparison between the pore
diameters deduced by Uno et al. (1998) and those of the authors. For interest, the measured results
between 0.065 mm and 0.85 mm have a better fit than those between 0.03 and 0.055 mm. This
indicates that the AIM had an overestimation of between 0.03 and 0.055 mm, and these results raise
a question that the pore diameter obtained from AIM is not a Poiseuille distribution.
VPM also evaluates the connectivity of the pore space. GIA provides not only the voxel number of
the sphere element, but also the spatial distribution with VPM. AIM can also provide the pore
diameter as an inner diameter of the pipe, but not the spatial distribution of the pore diameter. This
issue indicates that VPM has the great advantage of being able to estimate WRC. In fact, the
distribution curve in Figure 21 can provide a pore size distribution function (PDF) with respect to
the perforated pore diameter. PDF can also provide the saturation degree by summation of the voxel
and diameter of the sphere element. Figure 24 presents the WRC analyzed by VPM and PDF in this
study The WRC obtained from PDF was far from the results of VPM in terms of measured plots.
This issue poses the definition of pore diameter. As described in section 4.1, VPM considers
percolation using cluster labeling based on the connectivity of pore spaces. On the other hand, PDF
cannot provide the percolation property. Figure 24 concluded that a reasonable WRC can be
obtained from saturation degree and distribution of pore diameter concerned the percolation
property. Therefore, it is significantly useful that GIA and VPM can estimate the water retention
property based on the geometry of the pore structure without performing a WRC test.
**5. CONCLUSIONS**
In this study, a specimen of Toyoura sand was scanned using a micro-focused X-ray CT scanner,



and the 3D spatial distribution of a sphere element as the pore diameter (*d*) was visualized and
evaluated quantitatively by granulometric image analysis (GIA). The GIA was a useful image
analysis method to evaluate pore diameter in a 3-D CT image. Moreover, the voxel percolation
method (VPM) was newly developed in this study, and its validation was assessed by comparing
the analyzed WRC with measured results. The key conclusions are summarized as follows:
1) The size of a voxel affected the results of image analysis. When the cubic size of one voxel was 5 x
5 x 5 μm, the representative element volume (REV) to evaluate the physical property of grain
materials, which are similar to Toyoura sand, was at least 300 voxels for the evaluation of grain size,
porosity, surface area and perforated pore diameter. In particular, it was possible for the porosity
and surface area to evaluate the relative standard deviation less than 1 %;
2) Results of GIA show that the perforated pore diameter was less than the pore diameter from the air
intrusion method (AIM), and was less than 0.068 mm, meanwhile mostly similar pore diameters
were evaluated near 0.085 mm. Hence, AIM provided partially different pore diameters from the
results of GIA. This issue revealed that the pore diameter obtained from AIM was not Poiseuille
distribution.
3) AIT can estimate pore diameter as a diameter of a pipe and the occupation ratio; however, the
spatial information was not included and therefore, it was difficult to assess the water retention
curve (WRC) based on pore diameter and its occupation ratio. Meanwhile, the newly proposed
"voxel percolation method (VPM)" in this study can distinguish each sphere element by labeling the
number based on the diameter of the sphere element and scanning the continuous label. As a result,
the connectivity with complex pore spaces can be concerned and therefore, it was available for
VPM to provide the WRC close to the measured result.
4) It was concluded that the VPM was the better image analysis method, which could estimate the
water retention property due to drainage by percolation using cluster labeling (i.e. pore space) and
capillary pressure head based on the Young-Laplace equation, as long as image data of the pore
space was obtained by micro-focused X-ray CT scanner.
The second, third and fourth conclusions are based on the first conclusion. An appropriate
dimension for image analysis should be defined based on the particle diameter and voxel size. In the
case using the micro-focused X-ray CT scanner, the greater resolution required, the smaller the
sample that should be scanned. In future work, it will be necessary to verify the appropriate
dimension (i.e. REV) for several kinds of grains.





**ACKNOWLEDGEMENT**

This research was supported by a Grant-in-Aid for Scientific Research (C) No. 26420483. Authors thank Prof. Laurent Oxarango, who is an associate professor of University of Joseph Fourie, for his precious comment. We also thank Ms. Hitomi Miyahara, Chiaki Nagai and Yusaku Fujiki, who were a former bachelor and graduate school students of Kumamoto University, and Mr. Toru Yoshinaga and Mr. Takahiro Yoshinaga, who are technical staffs of X-Earth Center, for their sincere contribution to this research.

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





Figures

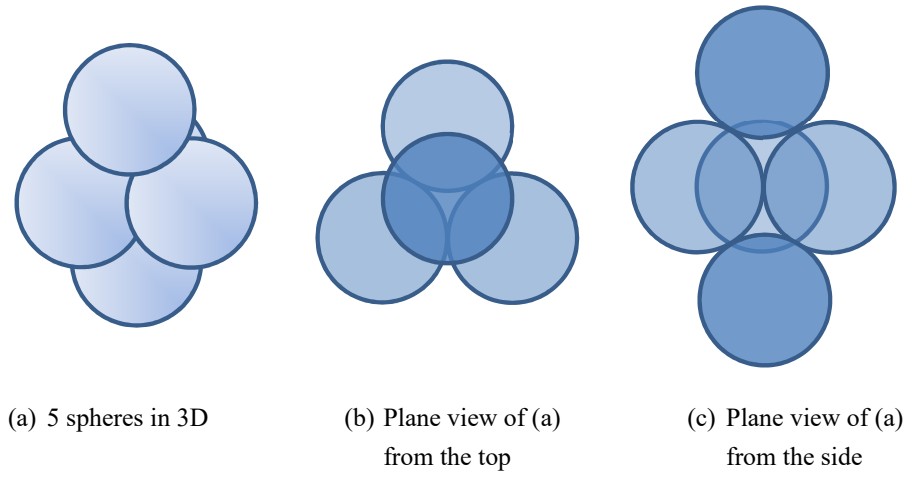

(a) 5 spheres in 3D

(b) Plane view of (a)
from the top

(c) Plane view of (a)
from the side

Figure 1: Illustration of spheres for pore surrounded particles

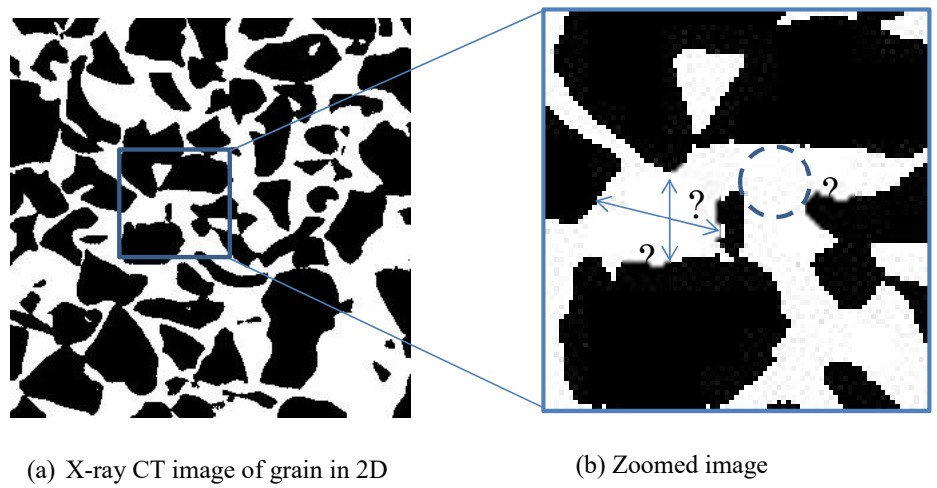

(a) X-ray CT image of grain in 2D

(b) Zoomed image

Figure 2 : Binary image of a pore in grains (white indicates pore space and black
indicates particles)



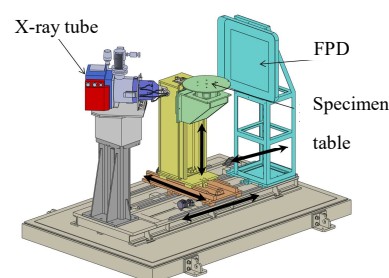

Figure 3: Illustration of micro-focused X-ray CT scanner

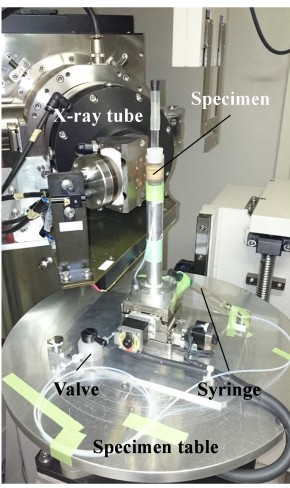

Figure 4: Photograph of a scan scene



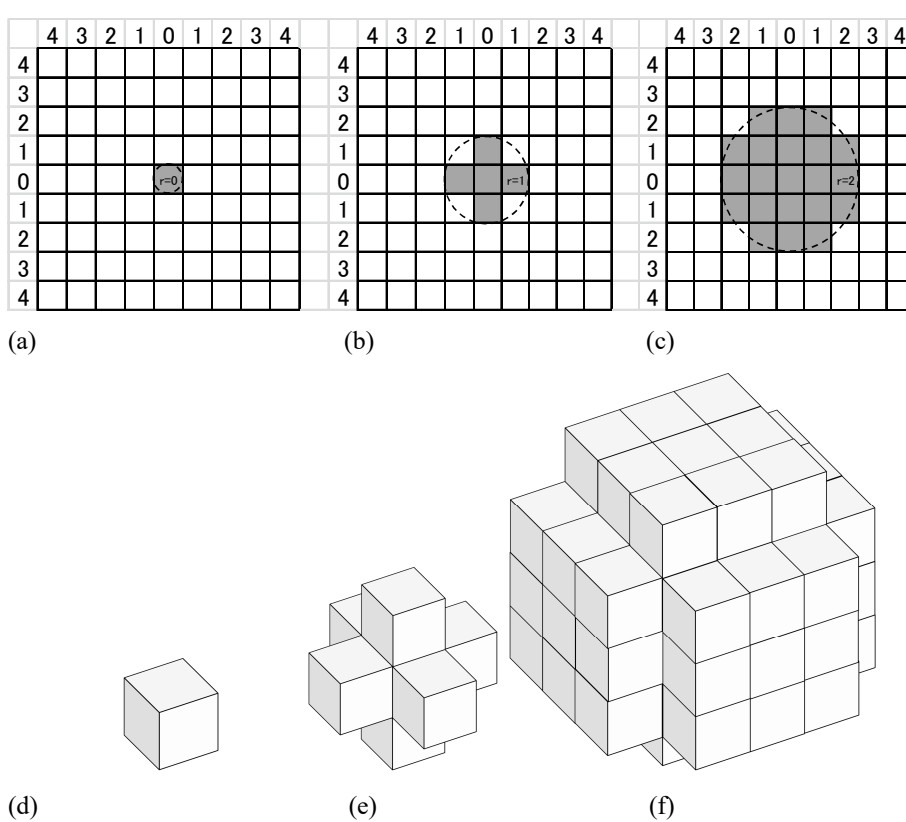

Figure 5: Illustration of sphere elements with different diameters in 2D and 3D views





(a) Case where particle shape is a square

(b) Case where particle shape is a circle

(c) Case where particle shape is a square

(d) Diamond shape

(e) Square shape or
45° rotated diamond in (d)

Figure 6: Explanation of features of GMI





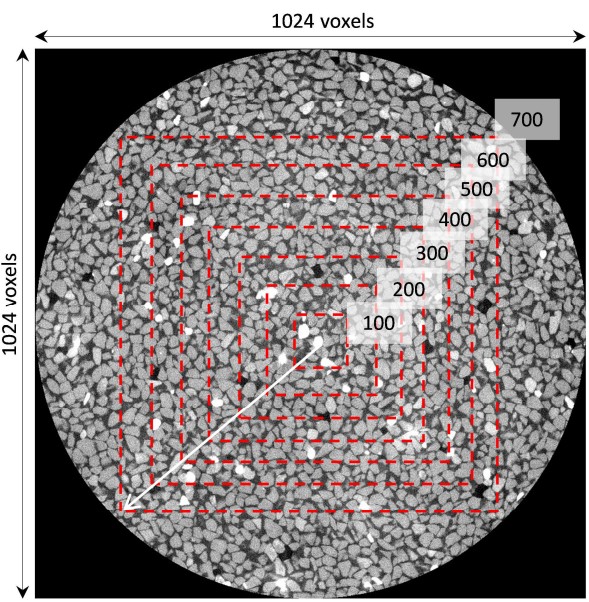

Figure 7(a): X-ray CT image of Toyoura sand with different regions of image analysis in 2D

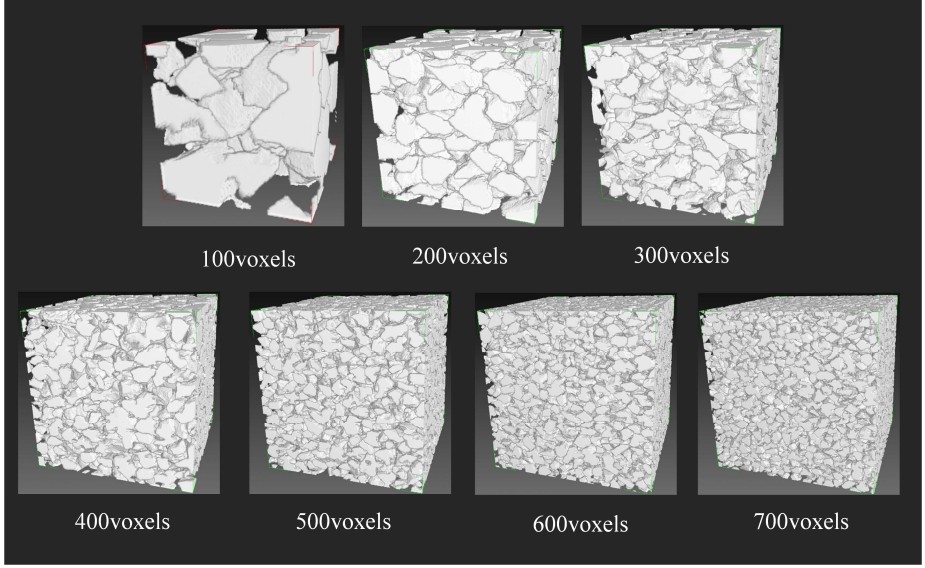

Figure 7(b): X-ray CT image of Toyoura sand with different regions of image analysis in 3D





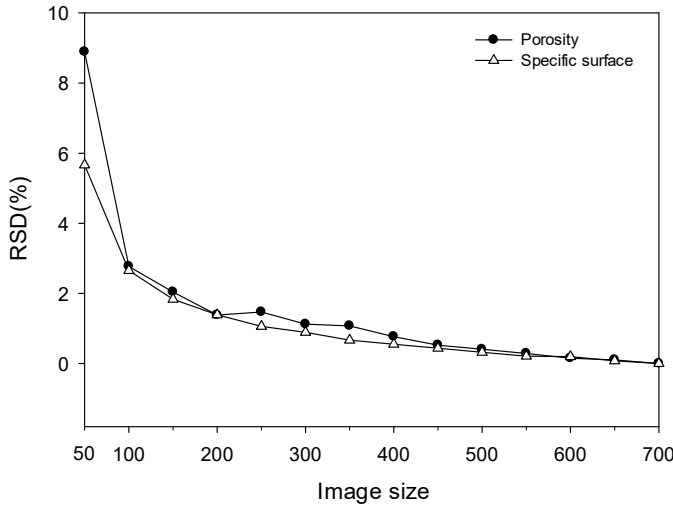

Figure 8: Relative standard deviation by changing the dimension of image analysis

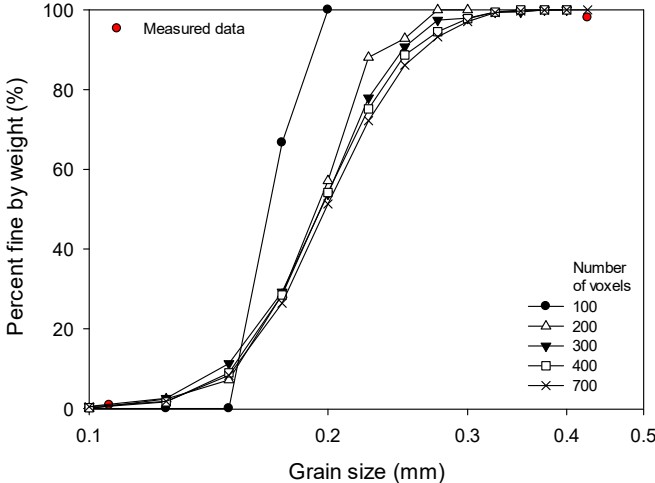

Figure 9: Grain size distribution curve obtained from image analysis



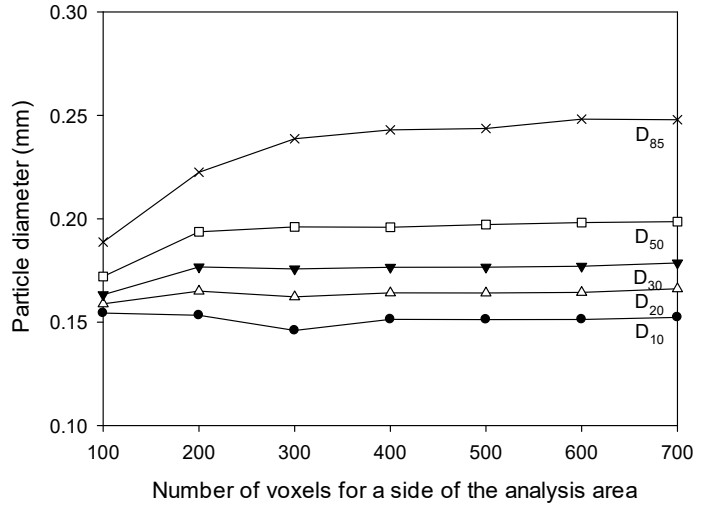

Figure 10: Grain level on each image size





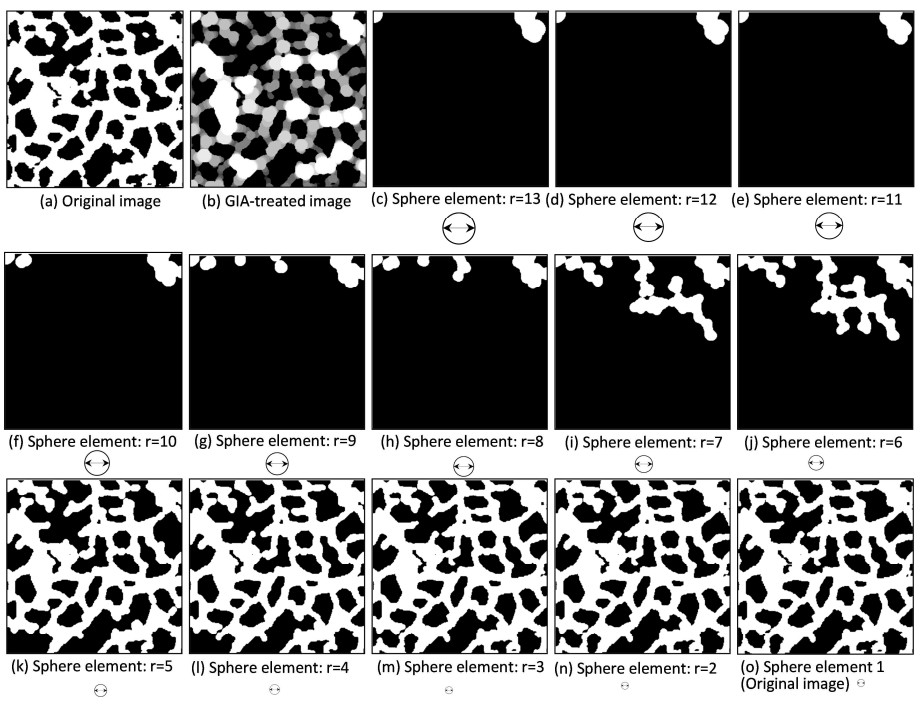

Figure 11: VPM analysis in grains





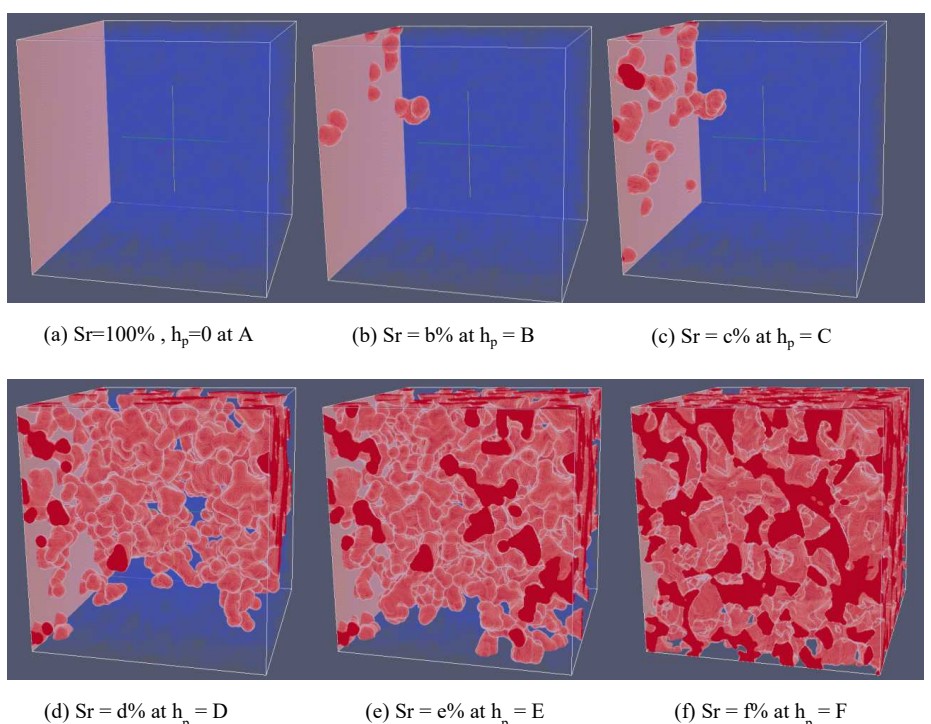

(a) Sr=100% , $h_p$=0 at A    (b) Sr = b% at $h_p$ = B    (c) Sr = c% at $h_p$ = C

(d) Sr = d% at $h_p$ = D    (e) Sr = e% at $h_p$ = E    (f) Sr = f% at $h_p$ = F

Figure 12: Voxel percolation method in grains

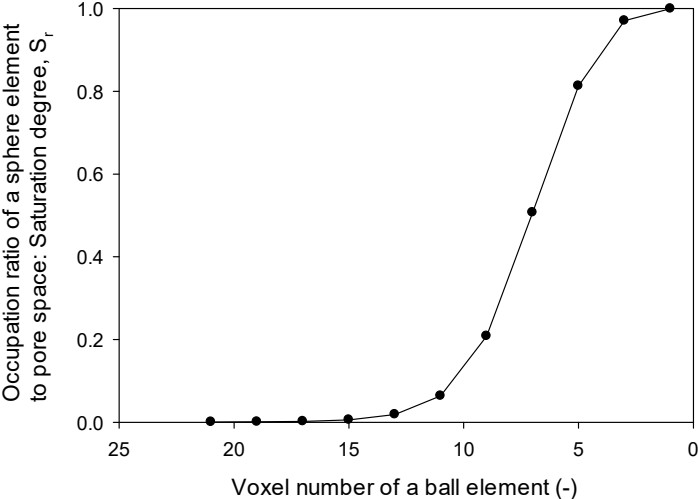

Figure 13: Profile of occupation ratio of the sphere element



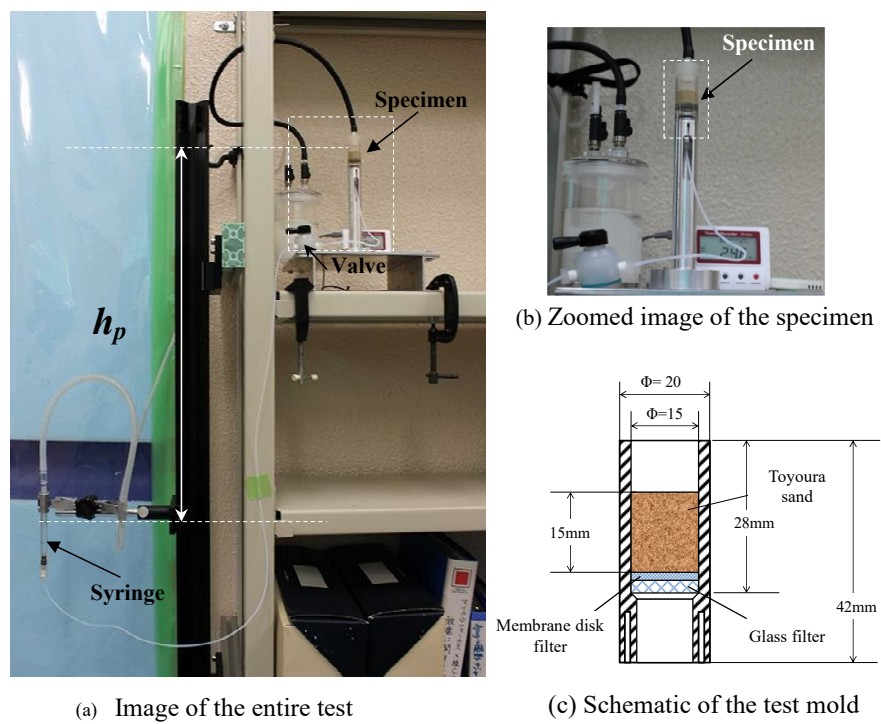

(a) Image of the entire test

(b) Zoomed image of the specimen

(c) Schematic of the test mold

Figure 14: Water retention test apparatus with elevation head method

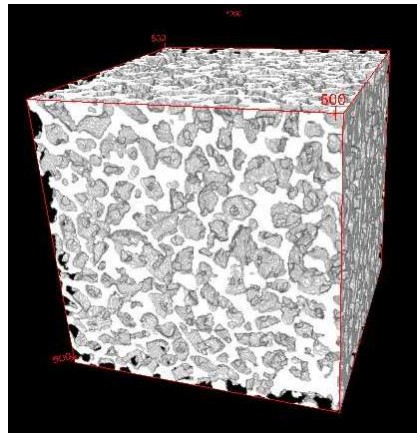

Binary image of pore space for Toyoura sand

Figure 15: X-ray CT image of pore in Toyoura sand in 3D





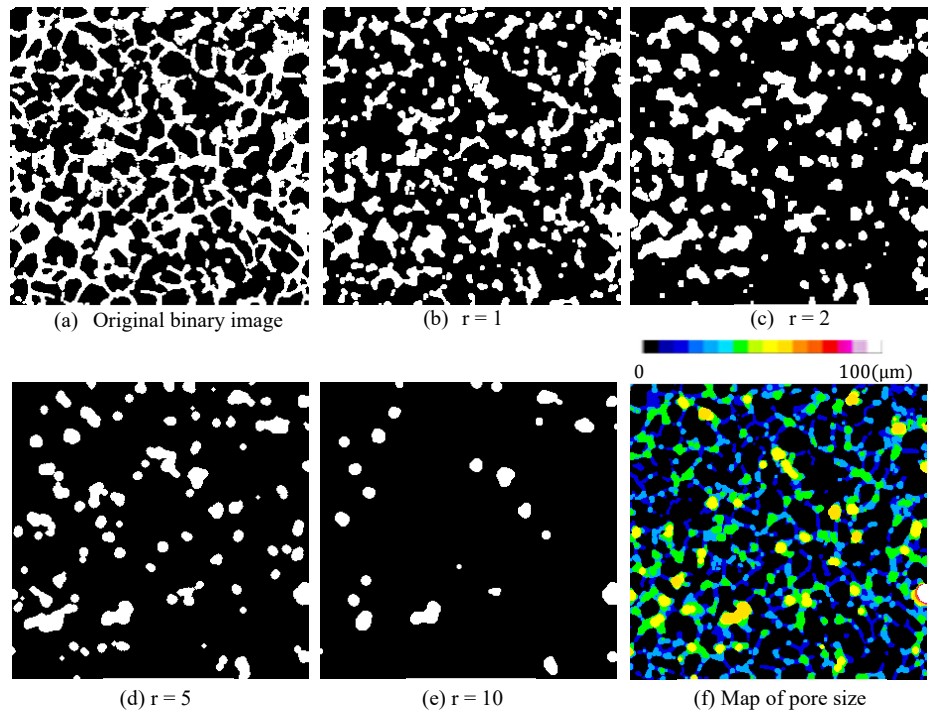

Figure 16(a), (b), (c), (d), (e), (f): X-ray CT images obtained from GMI

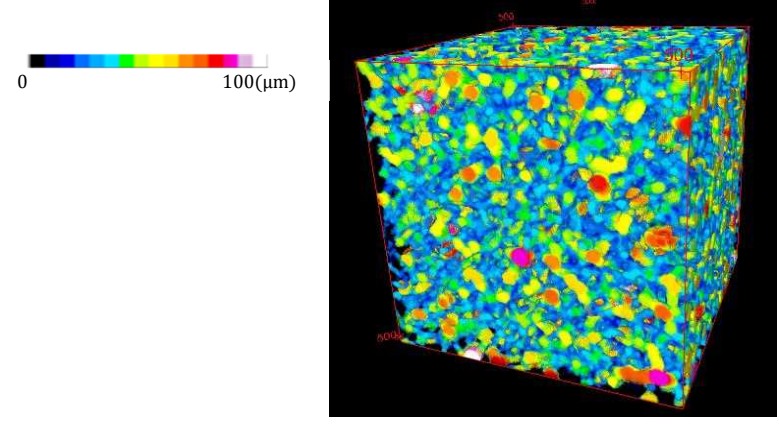

(b) 3D map of pore space for Toyoura sand

Figure 17: Distribution of perforated pore size in 3D





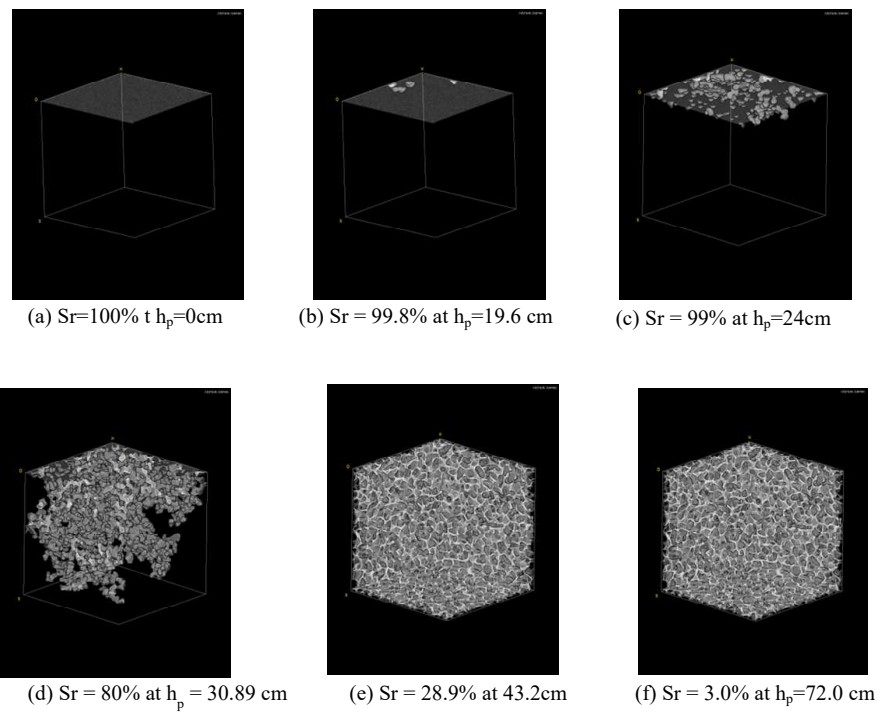

| (a) Sr=100% t $h_p$=0cm | (b) Sr = 99.8% at $h_p$=19.6 cm | (c) Sr = 99% at $h_p$=24cm |
| (d) Sr = 80% at $h_p$ = 30.89 cm | (e) Sr = 28.9% at $h_p$=43.2cm | (f) Sr = 3.0% at $h_p$=72.0 cm |

Figure 18: CT images of percolated pore space as the drainage process

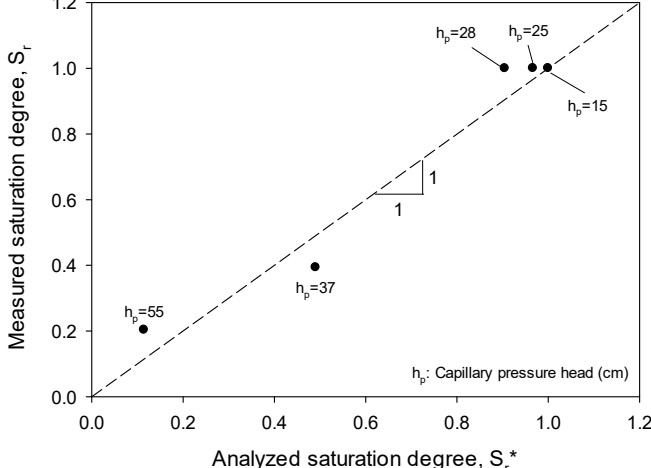

Figure 19: Comparison of saturation degree measured and analyzed at each capillary pressure





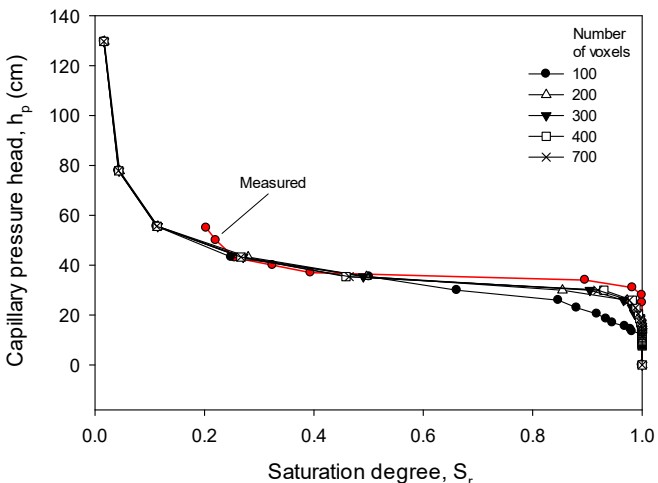

Figure 20: Water retention curves obtained from image analysis and experiment

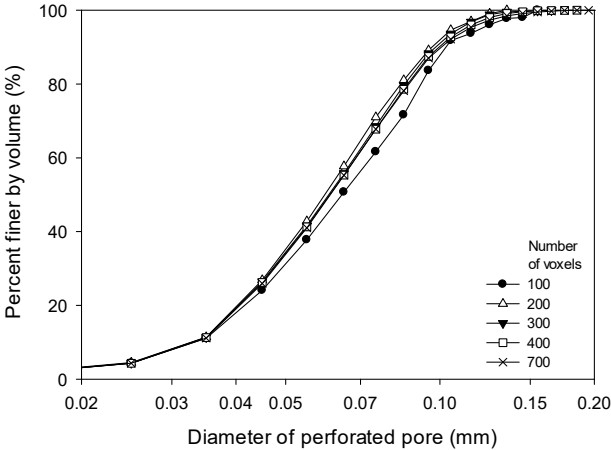

Figure 21: Perforated-pore size distribution for each image size





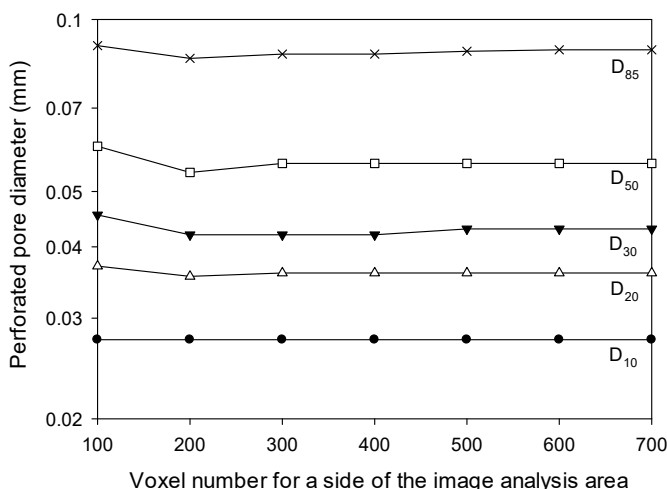

Figure 22: Pore size distribution curve in image size

Figure 22

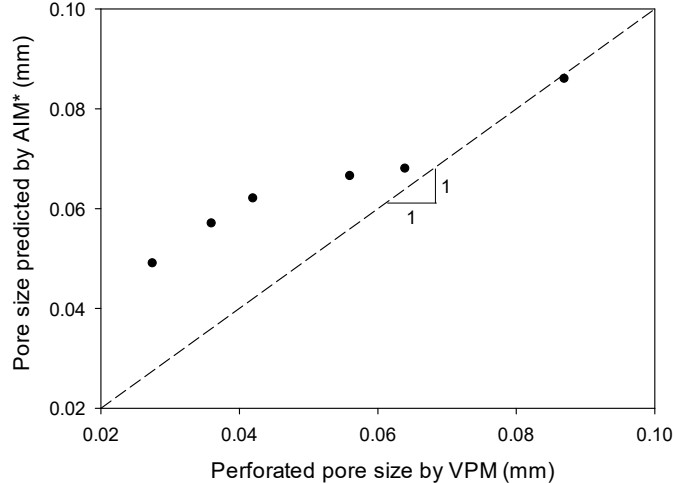

Figure 23: Comparison of pore size obtained from AIM and VPM




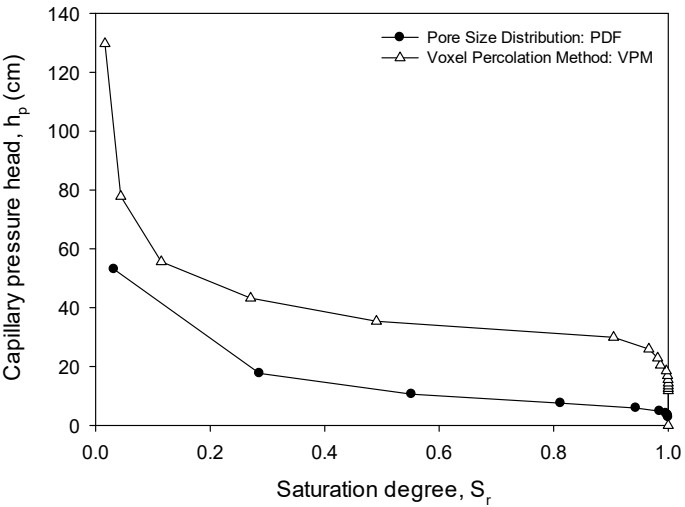

Figure 24: Water retention plots obtained from GMI and VPM



Table 1 Specifications of a micro-focused CT scanner installed at Kumamoto University

| | |
|---|---|
| Radiograph field vision | 400 mm, height 500 mm |
| Number of display voxels | 1024 x 1024 |
| Resolution | 4 μm |
| Cone bean scan | Normal, Offset, Half |
| X-ray beam thickness for plain beam | 0.2/0.4/0.6/1.0/2.0 mm |
| Voltage for x-ray generating | 240 kV (140W) maximum |
| Maximum weight for specimen table | 245 N |
| Flat panel detector | Effective pixel number: 2000 x 2000 |
| | Range of vision: 400mm x 400 mm |

Table 2 Scan conditions used in this study

| | |
|---|---|
| Power of voltage (kV) | 60 |
| Current (μA) | 200 |
| Number of views | 1500 |
| Number of integration treatments | 10 |
| Voxel dimension (μm) x, y, z | 5 x 5 x 5 |
| Number of voxels (x, y, z) | 1024 x 1024 x 1000 |

Table 3 Verification results of GMI

| | Voxel counts | | | Area by calculation | | | |
|---|---|---|---|---|---|---|---|
| | Solid | Pore space | Total area | equation | Solid | Pore space | Square error (SE)* |
| Circle | 31428 | 8572 | 40000 | $\pi r^2$ 100x100x3.1428 | 31428 | 8572 | 0 |
| Square | 19801 | 20199 | 40000 | L x L 140.716x140.716 | 19800.99266 | 20199.00734 | $5.393 \times 10^{-5}$ |
| Square Circle | 51229 | 28771 | 80000 | $\pi r^2$ + L x L | 51228.9927 | 28771.0073 | $5.329 \times 10^{-5}$ |

SE = (Voxel counts – Calculation)$^2$ for each case

Table 4 Results of REV analysis

| | Porosity,    Specific surface |
|---|---|
| RSD < 5 % | L > 900 μm (or 100 voxels) |
| RSD < 2 % | L > 1800 μm (or 160 voxels) |
| RSD < 1% | 1500 μm (or 300 voxels) |