# Peer review of "X-ray CT analysis of pore structure in sand"

_Solid Earth, 2016_

## Referee Comment (RC1) · Anonymous Referee #1 · 11 Mar 2016

Thank you very much for this paper. It match's perfect into the scope of the special issue in Solid Earth. General comments: The paper reports a XCT analysis of pore structure in a natural sand, compare proposed image analysis results with experimental water retention curve and discuss this in respect to REV of pore space. Mostly all details of developed and used methods are described (later follows some remarks and questions) and the application for this case study are demonstrated and discussed. Although the content and nice work presented in this paper, but it needs a major revision. But now looking more into details, first some remarks to the general structure of the presented paper and general questions: The paper overall is a bit "over-structured", some chapters contains short and large sub-chapters with sometimes similar content. Also several different kind of numerations lists (called step 1,2 . . . and other enumerations) hampers fluent reading and capturing the content. In the Introduction and also in the Discussion chapter, often a lot of literature is only numerate and it is not clear, who is doing what, except the reader knows the cited papers and authors. Better is

to rearrange citations and relate directly as reference for the field of research for example (specially p2,l4-6 and p2,l28-p3,l1). Same thing, but here it is better written in p13, l16-33). In general: many groups recently working in the field of pore scale imaging, pore morphology approaches and experiments, discussing REVs and a lot of work are published. I missing some recent citations (e.g. Hilfer et al., 2015 for multiphase REVs and length scales or very similar to you a work from Yang et al., Extraction of pore-morphology and capillary pressure curves from porous media from synchrotron-based tomography data, Scientific Reports, 5:10635, DOI: 10.1038/srep10635, 2015, and others ... The pore-morphology methods are meanwhile highly developed and include drainage and imbibition with different contact angels in complex geometries and realized in software packages (like GeoDict, see geodict.com). It would be nice to demonstrate more your approach of your combined image processing and pore-morphology methods, specially your combination of GIA and VPM looks interesting As we see in many publications, the development of interfacial area between different phases in multiphase flow (experiments and simulations) plays a fundamental rule in this topic. Multiphase flow processes deliver other REVs (and length scales) as "only" single phase or static systems (see Hilfer et al.). I missing here a discussion.

The material: you use Toyoura sand: on p4,l22-23 you give the number of dry density and porosity. What is the mineralogy of the sand? In Fig. 7a you can clearly see, there are different mineral phases. It would be also useful to have a grain size distribution curve (from sieving experiment) and if available some more sedimentological data about this sand (like grain shape analysis which have a lot of impact to pore space geometry) and compare this with your image analysis tools. In general: your image analysis implies that the pore structure (grain packing) is not changed over time (e.g. during compaction). You are sure, that in the water retention experiment compaction was not happened? You did only one experiment? Figures: The paper contains 24 figures! I think this is too much and should be reduced. E.g. figure 1 and figure 5 (illustration of sphere packings) is trivial and not necessary to show, also the figure 3 showing the CT scanner is not necessary. Also better would be a combination of fig. 4

and 14, as the experimental setup. The ordering of the figures should have rearranged. Better is to start with an CT reconstructed raw image (like fig. 7) and demonstrate, how the Toyoura sand is composed and represented as a XCT scan. Next figures can be fig. 15 as a segmented 3D image and 7b to demonstrate different boxel-sizes for determining the REVs. By the way: mark boxel sizes with 100**3 voxels, 200**3, voxels and so on … Fig. 2 is a good starting point (from image segmentation into your pore-morphology approach), but should arrange into a new order Fig. 6 is relative unclear to me. What are the colors? There is no description. Later more remarks for the appropriate chapter … (maybe delete) Fig. 8: it would be better to combine the real porosity/surface each boxel with absolute numbers and RSD in a different axis or as error bars. Fig. 9: shows grain distribution curve in respect to boxel sizes. Do you have only 2 points measured data for the grain size distribution curve of this sand or is this an accident in the figure? Fig. 10: better remove or combine with fig. 9 Fig. 11: nice to show how the VPM analysis works (very similar to other pore-morphology methods). 11 b): what are the gray levels (or colors) in the GIA treated image? Is that more or less the same as in fig. 17? If so, then please combine. Fig. 11 again: I think this is VPM in the segmented pore space not in grains! Fig. 12: same content like fig. 11, but in 3D. Maybe combine or remove. Fig. 13: is that a profile? I think this is a diagram, showing occupied pore space with VPM balls. It would be better to combine and demonstrate the outcome of VPM with Fig. 19. Fig. 14&15: see above Fig. 16: As I understood, this is a combination between GMI and GIA (shown in fig. 11). To reduce amount of figures, please only how fig. 17 as a final result of GMI, description how it works is the text and demonstrated principally in fig. 11 Fig. 18: Question: is that segmented air phase? How do you segment the air? This is not described and demonstrated in the paper. Maybe combine with description of image segmentation in general. As I understood, you use Otsu's method … Fig. 19: Do you have "only" 5 data points for the experiment, showed in fig. 18? I think you need more measurements between Sr=99% and Sr=28.9%. hp not corresponding to the hp in fig. 18 Fig. 20 shows a strong discrepancy between different boxel sizes and measure points (see remarks to the related

chapter) Fig. 21: is unclear to me. What is y-axis? Fig. 22: please remove. It can contribute in the text; content to REV and discussions Fig. 23: shows a discrepancy between AIM and VPM. This, I think needs more clear discussion in respect to a measured pore size distribution Fig. 24 also shows a clear a strong discrepancy between GMI and VPM (needs discussion) Text (please also see general remarks for structuring the paper and rearrangement of figures): P1, L27: I think you mean Young-Laplace law . . .. P2, L20: what means "pore dimensions"? P2, L23: MIT: I know it as MIP (mercury intrusion porosimetry) P3, L6-9: this is a bit unclear in the text. Please give an information why it is still under discussion. I think I know what you mean, but it needs to address more or delete this passage. P3, L11-12: often the language and formulations are not perfect, here repeating's like "this paper . . ." P3, L23: "this material has a uniform grain shape." That's NOT true! If you looking your images, there are different grain shapes! P4, L1-L13: can be shorter, because it is standard . . . P4, L23: please put some informations about mineral composition of the used sand P4, L26-27: do you use special ROI (region of interest) reconstruction method? P5, L3: "The pores has a . . ..", not one pore alone . . . P5-P6: please summarize and compact the text, often (e.g. P6, L21-22) repeats something . . . P6, L2: why only use 13 sphere elements? Is this the largest sphere radius? P7, L25: what are the real effect of rotation? If this used directly in the further image processing? As I understood, you use later only spheres (see L31) P8, formula 5: how do you calculate the surface? A kind of surface rendering or voxel surface count? Please give an information or cite something P8: there are often repeated text from pages before. (e.g. P8, L23ff). Please rewrite and compact P8, L21: as an idea: you can use cylindrical coordinates, so you are not limited to 700**3 voxels to determine REV. With cyl. oordinates you can use the full field of view of the scanned cylinder. For REV analyses, see my general remarks before. Table 2: what is the exposure/integration time for each projection? What hardware filter do you use to minimize beam hardening effect? Resulting voxels: you binned the projection image or use a ROI CT scheme? For all next pages, please follow my general remarks about the text and figures After rearrangement figures and text, it would be much easier to

follow the text content and discussion chapter

I would be happy, reviewing this paper after major revision.

---

## Referee Comment (RC2) · Anonymous Referee #2 · 21 Mar 2016

Reviewer's comments to the paper:

*X-ray CT analysis of pore structure in sand*
by Toshifumi Mukunoki et al.

This paper nicely fits the scope of this special issue of Solid Earth. I think that overall it is a welcome addition to the technical literature in the field, and its scientific content is quite good. However, in my opinion the current version of this paper requires some major revision, both in terms of the structure and most importantly in terms of the English, the quality of which is at times so poor that the reader doesn't understand the meaning of what the Authors are writing.

In the reviewer's opinion, the contribution under review cannot be published in its present form. The Authors should be encouraged to revise and resubmit their paper, but this will clearly require substantial modifications to the present version, and a new referee review.

**General remarks**

- Quality of the English: besides many typos and errors (too many to be listed herein: I'll just mention the word "retentively" often – but not always - used instead of "retention"), some statements are really hard to understand, e.g.,

    - *"This behavior indicates that mean pose size caused a capillary pressure head of 30-40 cm is mainly distributed. This behavior should be caused by sands with a value less than the uniform coefficient"*

    - *"Figure 24 concluded that a reasonable WRC can be obtained from saturation degree and distribution of pore diameter concerned the percolation property"*

    - *"it was possible for the porosity and surface area to evaluate the relative standard deviation less than 1%"*)

    - *"… requires that the water/oil flow in the soil quantitatively understand the pore structure"*

   I strongly suggest to the Authors to ask the help of a native English speaker.

- Figures: they are far too many (24!), and some of them are in fact quite useless – e.g., Fig. 1. Some other figures might be combined in one single figure.

- References: they are often cited in large groups, and it is not clear what is the criterion for citing those references rather than others. It would be better to cite fewer references, while making clear(er) what was studied/presented in each of them.

- Sections and subsections: they are far too many. The structure of the paper might be simpler, with fewer subsections.

**More specific remarks**

- It is stated in the introduction that *"In this paper, authors distinguish pore from pore structure"* (page 3, line 12). Yet, I haven't been able to find in the paper any clear definition neither of the former nor of the latter.

- At the end of the introduction, the Authors write that *"the evaluation of sand will be treated in this paper because it is natural material and has a uniform grain shape"*. This is just not true: Toyoura sand grains have not a "uniform" grain shape – whatever this means.

- Image segmentation (section 3.11): The very first step of image binarization is not discussed at all, which is weird – this is a crucial step in the analysis, because it affects all subsequent steps. They have used the image segmentation method developed by Otsu (1979). Are they aware that there are many other segmentation methods? The Authors simply inform the reader that "*… the Toyoura sand tested showed two distinct peaks in this study*", but they don't show any histogram of greylevel for a typical tomographic image – in my opinion they really should, this is mandatory.

- The explanation of the granulometric method (section 3.1.2, figure 5) is not very clear to me. I must confess I got lost – and yet I know a bit of image analysis…

- Also section 3.2 is rather obscure to me, I find Fig. 6 really not clear. The Authors are encouraged to substantially improve this section (and this figure), because this is an important part of their paper.

- In Fig. 9 (grain size distribution of Toyoura sand), the measured data are only two points. How is it possible?

- When commenting Fig. 23, the Authors state that "*the AIM had an overestimation of between 0.03 and 0.055 mm …*". The figure might in fact also indicate that VPM underestimates pore size in that range…

- Conclusions:
  - conclusion #1 is incorrect: the Authors cannot write that "*The size of a voxel affected the results of image analysis*", simply because they have studied only one voxel size. Of course I see what do they mean, and it is of course true that voxel size affects the results of image analysis. Affects, not affected: here is one of the numerous examples where a bad English can be misleading.
  - in conclusion #3, the statement "*This issue revealed that the pore diameter obtained from AIM was not Poiseuille*" deserves more explanations, I believe.
  - conclusion #4: that "*…the VPM was the better image analysis method distribution*" should be explained better.

**Further general remarks**

- On how many experiments is the paper based? Just one?

- The Authors are right when stating – at the very end of the paper – that "in future work, it will be necessary to verify the appropriate dimension (i.e. REV) for several kinds of grains". This is a very crucial issue: to what extent the results obtained in this study can be generalized to other granular materials and other resoultions/images? I believe the Authors should further comment on this.

---

## Author Comment (AC1) · 27 Apr 2016

Dear Editorial boards of Solid and Earth and Referee 1 and Referee 2

First of all, thank you very much for your kindness. Unfortunately, my area, Kumamoto was damaged by earthquake with M7.3 twice so we had to ask you to extend the due date of submitting revised manuscript. Recovering city and town has started so situation should be getting better, hopefully.

In this time, we submitted three files for each referee as follows: 1. Mukunoki_et_al_sand_void_paper second submission with revision record.pdf; 2. Mukunoki_et_al_sand_void_paper second submission without tracking.pdf; and, 3. Response to review results for referee1.pdf

Same files of the above item 1 and 2 are given to both referees. Detail explanation is included in the letter.

[Figure]

Thank you

Toshifumi Mukunoki

Please also note the supplement to this comment:
http://www.solid-earth-discuss.net/se-2016-16/se-2016-16-AC1-supplement.zip

––––––––––––––––––––––––––––––

---

## Author Comment (AC2) · 27 Apr 2016

Dear Editorial boards of Solid and Earth

First of all, thank you very much for your kindness. Unfortunately, my area, Kumamoto was damaged by earthquake with M7.3 twice so we had to ask you to extend the due date of submitting revised manuscript. Recovering city and town has started so situation should be getting better, hopefully.

In this time, we submitted three files for each referee as follows: 1. Mukunoki_et_al_sand_void_paper second submission with revision record.pdf; 2. Mukunoki_et_al_sand_void_paper second submission without tracking; and, 3. Response to review results for referee2

Same files of the above item 1 and 2 are given to both referees. Detail explanation is included in the letter.

[Figure]

Thank you

Toshifumi Mukunoki

Please also note the supplement to this comment:
http://www.solid-earth-discuss.net/se-2016-16/se-2016-16-AC2-supplement.zip